# Labeling on a Chip of Cellular Fibronectin and Matrix Metallopeptidase-9 in Human Serum

**DOI:** 10.3390/mi13101722

**Published:** 2022-10-12

**Authors:** Briliant Adhi Prabowo, Carole Sousa, Susana Cardoso, Paulo Freitas, Elisabete Fernandes

**Affiliations:** 1International Iberian Nanotechnology Laboratory (INL), 4715-330 Braga, Portugal; 2INESC-MN– Institute for Systems and Computer Engineering-Microsystems and Nanotechnologies,1000-029 Lisbon, Portugal

**Keywords:** microfluidic chip, cellular fibronectin, MMP9, serum, magnetic nanoparticle, magnetoresistive, biosensor

## Abstract

We present a microfluidic chip for protein labeling in the human serum-based matrix. Serum is a complex sample matrix that contains a variety of proteins, and a matrix is used in many clinical tests. In this study, the device performance was tested using commercial serum samples from healthy donors spiked with the following target proteins: cellular fibronectin (c-Fn) and matrix metallopeptidase 9 (MMP9). The microfluidic molds were fabricated using micro milling on acrylic and using stereolithography (SLA) three-dimensional (3D) printing for an alternative method and comparison. A simple quality control was performed for both fabrication mold methods to inspect the channel height of the chip that plays a critical role in the labeling process. The fabricated microfluidic chip shows a good reproducibility and repeatability of the performance for the optimized channel height of 150 µm. The spiked proteins of c-Fn and MMP9 in the human serum-based matrix, were successfully labeled by the functionalized magnetic nanoparticles (MNPs). The biomarker labeling occurring in the serum was compared using a simple matrix sample: phosphate buffer. The measured signals obtained by using a magnetoresistive (MR) biochip platform showed that the labeling using the proposed microfluidic chip is in good agreement for both matrixes, i.e., the analytical performance (sensitivity) obtained with the serum, near the relevant cutoff values, is within the uncertainty of the measurements obtained with a simple and more controlled matrix: phosphate buffer. This finding is promising for stroke patient stratification where these biomarkers are found at high concentrations in the serum.

## 1. Introduction

Detecting specific biomarkers in physiological body fluids, such as blood, serum, plasma, sweat, and urine is a challenge in the development of biosensors [1,2,3,4]. Even more, this is highlighted when the sample preparation takes a considerable amount of time and decreases the diagnostic tool’s effectiveness. Analytical parameters such as selectivity and specificity are a further challenge for the low concentrations of biomarkers. For example, in emergency medicine, the stratification of ischemic stroke patients should occur in the shortest amount of time possible, because of the narrow time window for thrombolytic therapy [5,6,7,8]. Nevertheless, human blood is a complex sample matrix that can be difficult to access the relevant stroke biomarkers, such as cellular fibronectin (c-Fn) and matrix metallopeptidase 9 (MMP9) [6,9,10]—strongly associated with ischemic stroke screening in clinical studies for more than five decades [11,12]. When the obstacles to sample a matrix complexity cannot be overcome, a sample preparation method preceding the detection is needed [3,13].

Microfluidics spark much attention from scientists in sample pre-treatments, such as the sample preparation of devices for complex biological matrices, particle sorters, labeling processes, mixers, and flow manipulation from liquid mediums [13,14,15,16,17,18,19,20]. The conventional fabrication of microfluidic devices uses the casting polymer method on a mold as a master pattern that is fabricated using a costly lithography process [2]. Low-cost fabrication molds were introduced using micro-milling to engrave acrylic materials, simplifying the laborious lithography process and significantly cutting the fabrication cost [21,22]. Moreover, the straightforward fabrication process that can be accomplished outside the cleanroom facilities, acrylic is an incredibly cheap material, transparent and robust for the master pattern of microfluidics, with a resolution down to 100 µm [21,23]. Nevertheless, reproducibility and a low throughput yield are still the main issues of the micro-milling fabrication method.

The 3D printing process offers better reproducibility and a high throughput production of the microstructure for microfluidics [24,25]. Several methods for 3D printing techniques, such as selective laser sintering, the extrusion method, inkjet printing, and stereolithography (SLA), offer a broad spectrum in terms of cost, resolution, roughness, stiffness, and additive materials [26,27,28].

Herein, we demonstrate and compare the reproducibility and repeatability of a microfluidic chip for the magnetic labeling of protein biomarkers using molds fabricated using micro-milling [29] and 3D printing with a simple quality control. For that, we used two relevant stroke biomarkers, c-Fn, and MMP9. According to the literature, the simultaneous detection of these biomarkers can provide 87% prediction specificity for HT risk (the main side-effect of the thrombolytic treatment in acute cases of stroke) [6]. To our knowledge, the development of a microfluidic chip for magnetic labeling of relevant stroke biomarkers using a human serum-based matrix is firstly presented. Moreover, the performance is compared when labeling occurs in a simple matrix: phosphate buffer. For the quantification of biomarkers, a magnetoresistive (MR) biochip platform was used, that validates the performance of the target labeling of c-Fn and MMP9, occurring on the MF chip.

## 2. Materials and Methods

### 2.1. Microfluidics Mold Fabrications

Microfluidic designs were drawn using AutoCAD 2013 (Autodesk Inc., San Francisco, CA, USA). Next, the design was fabricated using both micro-milling and SLA 3D printing. The mold fabrication, using micro-milling computer numerical control (CNC), was simulated using ArtCAM software for mapping the sequence of the engraving and cutting (Figure 1A), the process is described in detail in [29].

For the micro-milling fabrication, two approaches were used: optimized and non-optimized micro-milling. The optimizations were set up in the ArtCam software. The optimized method used the parameters of the milling step equal to half of the tool diameter. In this study, the milling tool with a 0.8 mm size was used, therefore the milling step of 0.4 mm was used. While for the non-optimized micro-milling, the milling step was 0.8 mm. The optimized fabrication created a smoother structure due to the low resolution of the milling with a total fabrication process of more than 12 h. Conversely, the non-optimized micro-milling resulted in a rough structure with a faster fabrication process of around 6 h. The limitation of the long fabrication process for a single mold structure in micro-milling can be overcome by 3D printing, with a comparable fabrication time to produce several structures simultaneously (up to six molds).

The 3D design from AutoCAD was exported into.stl files and printed using an SLA 3D printer (Formlabs, Somerville, MA, USA) with the smallest resolution setting of 25 µm. The SLA 3D printer used a 1 L cartridge of resin (grey v4, Formlabs, Somerville, MA, USA). The printing orientation of the mold was 45° with a full raft support (Figure 1B). Once the printing process was completed, the mold structure was immersed in isopropyl alcohol (IPA) under an ultrasonic bath (Formlabs, Somerville, MA, USA) for 10 min, followed by resin curing (Formlabs, UK) at 65 °C, for 1 h under the ultraviolet light. Next, the printed structures were stored in an oven at 65 °C for 8 hr. The fabricated molds from both the micro-milling and 3D printing (Figure 1C,D) were then washed using DI water and IPA before the PDMS casting.

### 2.2. PDMS Casting and the Microfluidic Inspection

Polydimethylsiloxane (PDMS) elastomer (Dowsil™ 184 Silicone Elastomer, Dow Chemical, Midland, MI, USA), with a cured ratio solution of 10:1 was stirred and mixed (3 min, room temperature/R.T.). Next, the elastomer solution was degassed in the vacuum to remove the bubbles until the transparent color was achieved. Subsequently, the elastomer was cast into the mold and then baked in the oven (2 h, 65 °C). The hardened elastomer peeled off from the mold. A microscope slide with a size of 75 mm × 25 mm × 1 mm (Epredia™ Microscope Slides, Thermo Scientific, Waltham, MA, USA) was cleaned in DI water and isopropyl alcohol (IPA) (Merck, Darmstadt, Germany) and used as a substrate. Both the patterned elastomer surface and the microscope slides were exposed under plasma O_2_ (Expanded Plasma Cleaner PDC-002, Harrick Plasma, Ithaca, NY, USA) for 40 s. The patterned elastomer was flipped to the glass substrate and bonded with uniform pressure to all of the surfaces. The microfluidic channel was functionalized using poly(dimethylsiloxane-b-ethylene oxide) (PDMS-b-PEO) as described in detail in [27]. A permanent magnet with a disc shape (d = 15 mm, supermagnete, Webcraft GmbH, Gottmadingen, Germany) was positioned under the concentrated chamber to collect the labeled biomarkers before the MR sensor measurement.

For the flow test (Figure 1E), 100 µL (*V_T_*) of phosphate buffer (PB) was mixed with 2 µL of magnetic nanoparticles (MNPs) (d = 250 µm, Nanomag, Micromod Partikeltechnologie GmbH, Rostock, Germany). The timer (*t*) is counted from the first drop and stops until the liquid reaches the outlet. The rest of the liquid in the inlet was pipetted and calculated for the rest of the volume (*Vi*). The flow rate (*FR*) of the channel can be estimated by the following formula:(1)$$FR=\frac{V_{T}-V_{i}}{t}$$

For the channel inspection and the quality control (Figure 2), the freshly peeled elastomer was sliced across the channel into thin pieces. Next, the sliced chips were attached to the microscope slide and inspected under the microscope (Wide-Field Upright, Nikon–Ni-E, Nikon, Tokyo, Japan). Figure 2 depicted the channel heights from the cast elastomer from the fabricated molds.

### 2.3. MNPs Functionalization

Two μL of streptavidin-conjugated MNPs (d = 250, concentration = 4.9 × 10^11^ mL^−1^, Nanomag, Micromod Partikeltechnologie GmbH, Rostock, Germany) were incubated with 50 μg/mL of biotinylated polyclonal antibodies (Immunostep, Salamanca, Spain) of c-Fn or MMP9 for 1 h. Next, the bovine serum albumin (BSA) (Sigma-Aldrich, Waltham, MA, USA, Merck KGaA, Darmstadt, Germany) in 5% phosphate buffer (PB) was used for the MNP surface blocking during the 1 h incubation. Then, the MNPs for labeling c-Fn were diluted four times (1:4) and the MNPs for MMP9 were undiluted (1:1). The MNPs were mixed with the serum-based matrices spiked with the interested biomarkers in the inlet microfluidic chip area, in a total volume of 100 μL. The detail of the MNP functionalization is reported elsewhere [29].

### 2.4. MR Sensor and Its Functionalization

The MR biochip is composed of an array of 30 U-shaped spin-valve (SV) sensors and was microfabricated, as reported previously [30,31,32]. The gold thin layer (12.9 × 35.4 μm^2^) above the SV sensors was treated with a heterobifunctional surface linker: 1 mg/mL of sulfosuccinimidyl 6-[3′-(2-pyridyldithio)propionamido] hexanoate (sulfo-LC-SPDP, Thermo Scientific, Waltham, MA, USA) in a PB for 20 min. Then, 1 µL of the monoclonal antibodies (Abcam, Cambridge, UK) at 250 µL/mL, made contact with the sensor surface of 15 SV sensors for 2 h at room temperature (RT), followed by a blocking step with 1% of the BSA. The remaining 15 SV sensors were used as negative control sensors and contacted with 5% of the BSA.

### 2.5. Spiked Biomarkers in the Human Serum-Based Matrix

Human serum from male AB and purchased from Sigma Aldrich (Merck, Darmstadt, Germany) was used untreated. Next, the target proteins of c-Fn and MMP 9 were prepared at different concentrations around the clinical cutoff value of 3.4 µg/mL and 300 ng/mL, for c-Fn and MMP9, respectively (in a total volume of 100 µL of serum). Therefore, c-Fn was prepared in several concentrations of 1, 4, and 10 µg/mL; while MMP9 was prepared in concentrations of 100, 300, and 1000 ng/mL. In this study, the MNPs were used at a dilution ratio of 1:4 for c-Fn and undiluted for MMP9. The human serum samples, spiked with the respective biomarkers, were loaded into the microfluidic chip together with the functionalized MNPs and followed the protocol described in [29]. The result of the labeling process: the MNP-labeled specific biomarkers were quantified at the MR biochip platform. The mechanism of the signal acquisition from the MR sensor is described in Figure 3A,B. Briefly, a measurement is composed of three steps: (1) The baseline: the baseline is acquired by passing a PB for 5 min over the MR sensor surfaces, which are immobilized with the monoclonal antibodies. (2) The saturation: the MNP complexes, i.e., the MNP plus the specific target captured by the microfluidic chip, are inserted into the biosensor flow cell. Five min later, the magnetic focusing is applied to attract the MNPs to the top of the MR sensor surface to distribute the MNPs uniformly and to allow opportunities for the monoclonal antibodies to capture the MNP complexes. (3) The washing: after the saturation period of up to 25 min, the sensor surfaces are washed with a PB-T buffer (phosphate buffer with 0.05% Tween^®^20) until the signal is stable. In this step, most of the non-specific binding can be released from the sensor surface, leaving only the specific binding. Finally, the differential signal between the baseline and the saturation level, after the wash, was recorded as the binding signal.

### 2.6. Statistical Analysis

The MR biochip was composed of an array of 30 U-shaped SV sensors, where 15 were used for target binding, and the other 15 for the negative control. The signals were collected from at least 10 sensors for the target and negative controls, respectively.

The measured voltage signal in each sensing area was normalized using the following formula:V_NORM_ = ∆V/V_SENSOR_(2)
where V_NORM_ is the normalized signal value and unitless, ∆V is the delta between the average of the baseline and binding signal, and V_SENSOR_ is the average of the baseline (sensor output).

The average signals of the active sensors in a measurement batch were calculated using the propagation of errors approach, where each average and standard deviation contribute to the calculated signal value. The curve fittings were obtained using the Hillslope model, with a 95% confidence level of the calibration curve, using 1000 data points, ANOVA using a maximum of 400 iterations. The signal level of the detection limit (Y_LOD_) was calculated using the formula from [33]: Y_LOD_ = 1.645 SD_C0_ + 1.645 SD_C1_(3)
where SD_C0_ and SD_C1_ are the standard deviations from the blank measurement and the smallest concentration, respectively.

## 3. Results and Discussion

### 3.1. Repeatability of the Microfluidics

The molds fabricated by the CNC and 3D printer were compared to a reference mold previously fabricated in other published study on the CNC [29]. The parameters checked for the quality control (QC) of the microfluidic chips produced from those different fabricated molds, are summarized in Table 1. The molds with an acceptable QC were used for the PDMS casting to obtain the final microfluidic chips, which were tested against the targets of interest (c-Fn and MMP9). Following the labeling on a chip, the levels of the proteins were quantified using an MR biochip platform to demonstrate the reproducibility of the molds, and the respective repeatability of the measurements (Figure 4). The magnitude of the MR sensor signals is in a comparable range to the reference signal (light yellow region) obtained from our previous study [29]. In contrast, the microfluidic chips with a higher channel thickness show that the signals were significantly lower than those that passed the QC.

We provide, in Figure 5, an illustration to explain the hypothesis behind the obtained results. The microfluidic with the optimum channel height (Figure 5A) shows a filtration mechanism by the MNP cluster in the concentrated chamber. The MNP clusters were concentrated around the magnetic field of the permanent magnet, to capture the biomarkers that flow with the sample medium. Moreover, the microfluidic chips with a higher channel height decreased the filtration efficiency of the functionalized MNP clusters. The MNPs tend to be clustered in the substrate due to the magnetic field from the magnet, while the channel gap above potentially passes the biomarkers to the sponge. Moreover, the high channel heights enhanced the sponge absorption due to the increased surface contact with the medium. Although the idea of making the channel height as small as possible can lead to an effective filtration, the longest absorption time is due to the light contact of the sponge surface with the medium.

In addition to the channel height, we believe that other variables can also influence the effectiveness of the MNP filtration for the proposed microfluidic chip. First, the MNP concentration is crucial to the dynamic range that can be tuned to fit the clinical cutoff value [27,28]. It is essential to obtain enough MNPs to cluster in the reaction chamber. Second, is the preference of the magnet, i.e., the shape and the strength of the permanent magnet contribute to the filtration of the biomarkers. Weak magnetic fields may allow some functionalized MNPs to be absorbed into the sponge.

### 3.2. Protein Labeling Using a Human Serum-Based Matrix

Stroke biomarkers were spiked in human serum-based matrices at the following concentrations: c-Fn at concentrations of 1, 4, and 10 µg/mL, and MMP9 at concentrations of 100, 300, and 1000 ng/mL. These concentrations were prepared to understand if the technology (microfluidic chips and MR biochip platform) can be used to detect the biomarkers near the relevant clinical cutoff values defined for stroke patient stratification, at 3.4 µg/mL and 140 ng/mL for c-Fn and MMP9, respectively [12,29]. To obtain the dynamic range that fit the clinical cutoff value, the undiluted (1:1) and the diluted (1:4) MNPs were used for the labeling of c-Fn and MMP9, respectively [29].

The quantification of c-Fn and MMP9 spiked in the human serum is presented in Figure 6A,B. The measurements obtained using a complex sample matrix (serum) were cross-validated against the measurements obtained in the proof-of-concept of the microfluidic device using biomarkers spiked in a PB. The results showed that the measurements performed using a complex sample matrix were in a good agreement with the calibration curves of c-Fn and MMP9 labeled in the buffer. The standard deviation of the measurements obtained in the serum, however, was slightly higher. The estimated limit-of-detection (LOD) were 205.14 ng/mL and 37.5 ng/mL for c-Fn and MMP9 in the human serum-based matrix, respectively. These values are higher compared to the measurements obtained in a simple matrix: the buffers, that were 54.6 ng/mL and 11.5 ng/mL for c-Fn and MMP9, respectively. This can be explained by the complex matrix of the serum, derived from the whole blood that contains several proteins that can increase the noise of the biosensors and lead to non-specific binding events. It was reported that in the human serum, there are around 325 distinct proteins [34]. Albumin or immunoglobulins G are major proteins that can make difficult the interaction between the specific antibodies attached to the MNPs and the interested biomarkers. Nevertheless, using the proposed microfluidic chip, the filtration mechanism described above (Figure 5), by the MNPs occurs gradually, and it is sorted in a spotted area. It is utterly different from a conventional labeling method occurring in an Eppendorf™ tube that takes up to 1 h (because of the static incubation).

## 4. Conclusions

We have demonstrated a functional microfluidic chip for protein labeling in the human serum. The microfluidic molds were fabricated using the micro-milling method and a 3D printer, with the optimized reproducibility for a channel height of 150 µm. The repeatability of the microfluidic chips was successfully demonstrated by performing measurements using an MR biochip platform for several days—a proof-of-concept for the quality control of the microfluidic chip production. The channel height plays the most crucial role in the labeling performance because of the magnetic field that exists from the bottom of the chip. The optimization of the channel shapes and the different pattern of magnetic fluxes are potentially explored for future work. Finally, the MNP labeling of c-Fn and MMP9 in the human serum can be performed in the proposed devices, with a comparable performance for labeling the biomarkers in the buffer. The proposed method is feasible for the performance of clinical studies involving the tested biomarkers.

## Figures and Tables

**Figure 1 micromachines-13-01722-f001:**
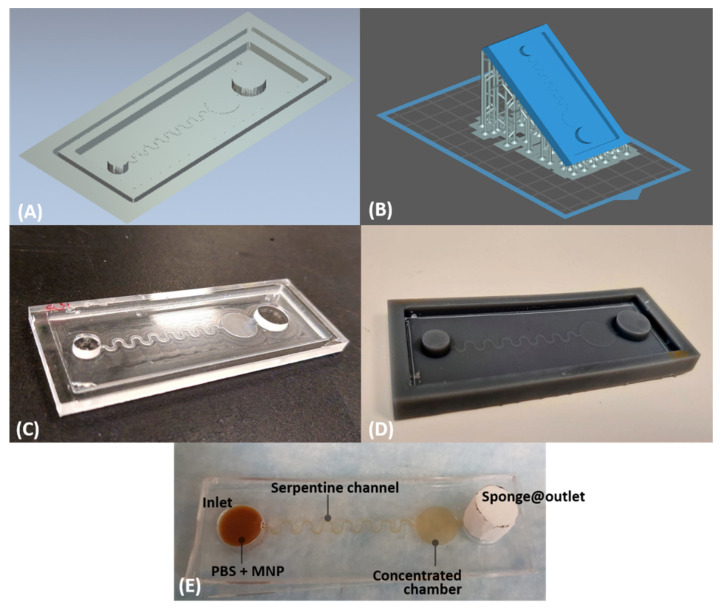
(**A**) Simulated micro-milling fabrication of the microfluidic mold in acrylic. (**B**) 3D model of the mold with 45° printing orientation using a raft support. Fabricated mold using (**C**) micro-milling and (**D**) SLA 3D printing. (**E**) Fabricated microfluidic chip for the test flow. The dimension of the microfluidic chip: total length = 75 mm; width = 25 mm; diameter of the inlet = 7.5 mm, diameter of the concentrated chamber and the outlet = 10 mm; height of the inlet and the outlet = 3 mm; height of the channel and the concentrated chamber = 250 µm; width of the serpentine channel = 250 µm; total length of the serpentine channel = 88 mm.

**Figure 2 micromachines-13-01722-f002:**
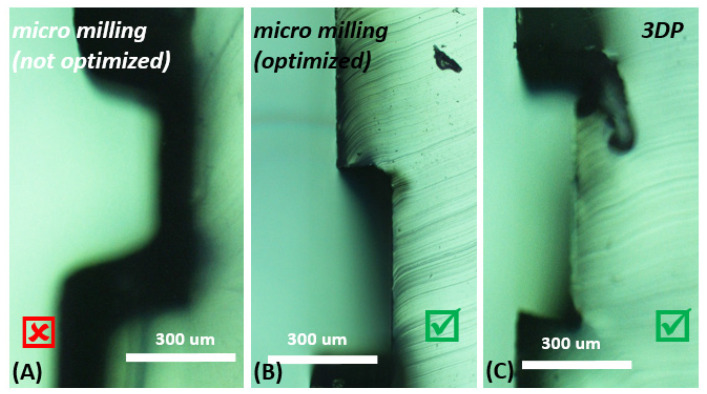
Inspection of the microfluidic channel from the cast elastomer under a wide-field upright microscope. (**A**) Microfluidic chip from the micro-milling mold (not optimized). It shows the channel height of around 276 µm, which does not pass the quality control. Microfluidic chip, which passes the quality control with a channel height ~150 µm; fabricated from the mold using (**B**) the optimized micro-milling and (**C**) SLA 3D printing.

**Figure 3 micromachines-13-01722-f003:**
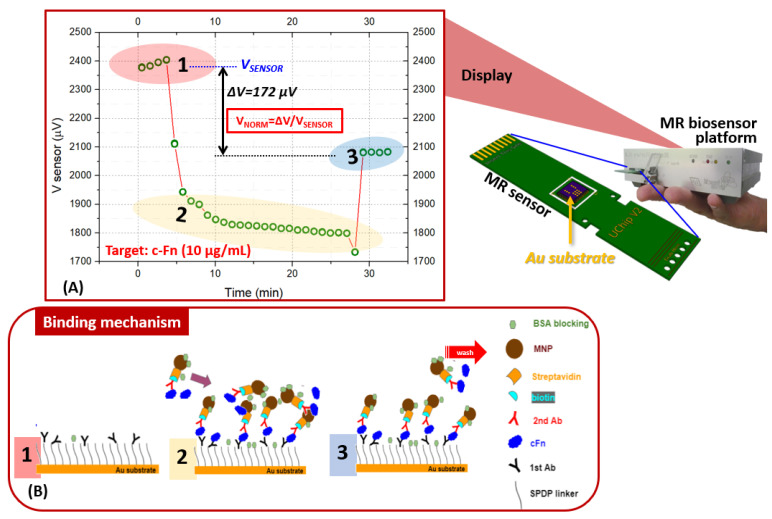
Illustration of: (**A**) a representative image of the binding signal measurement obtained for c-Fn using a portable MR biochip as a sensing element. (**B**) the sandwich assay strategy occurring on the gold surface of a MR sensor.

**Figure 4 micromachines-13-01722-f004:**
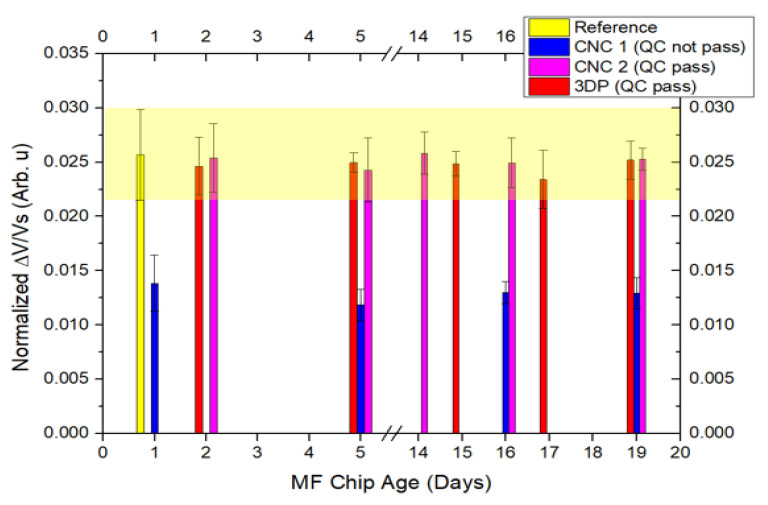
Reproducibility of the molds fabricated by the micro-milling computer numerical control (CNC) and k3D printer (3DP), and the respective repeatability of the measurements up to 19 days. The reference signal (yellow color) was referred from our previous work [29], and used here for comparison—a MF chip was obtained from a mold fabricated using CNC, and used for labeling c-Fn (10 µg/mL) in buffer.

**Figure 5 micromachines-13-01722-f005:**
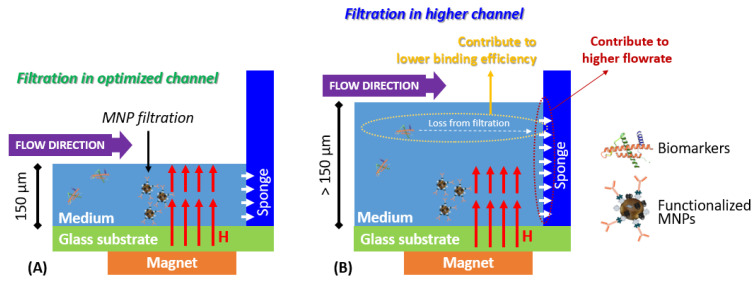
The critical role of the channel height of the microfluidic chip is to capture the target protein. (**A**) An suitable protein labeling in an optimized channel. (**B**) A higher channel reduces the protein labeling efficiency and increases the contact surface of the medium with the sponges.

**Figure 6 micromachines-13-01722-f006:**
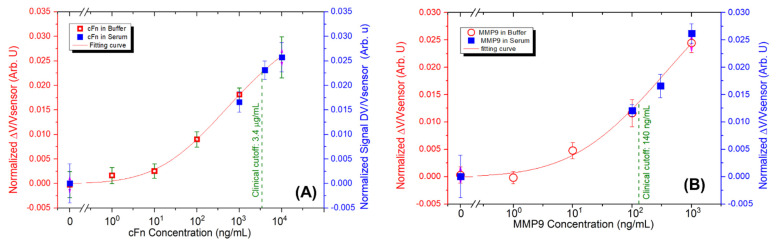
Quantification of c-Fn (**A**) and MMP9 (**B**) in the human serum-based matrix (blue) near the relevant clinical cutoff regions. Measurements were cross-validated with the calibration curve obtained using the biomarkers spiked in a phosphate buffer (red). The green dash depicts the clinical cutoff value for each biomarker.

**Table 1 micromachines-13-01722-t001:** Inspection and quality control test: microfluidics chips from the different fabricated molds.

	Reference Mold	CNC	Optimized CNC	3D Printer
Volume (serpentine + chamber)	25 µL	46 µL	25 µL	25 µL
Channel height	150 µm	276 µm	150 µm	150 µm
Quality control	-	Failed	Passed	Passed

## Data Availability

Not applicable.

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
