# Peer review of "Labeling on a Chip of Cellular Fibronectin and Matrix Metallopeptidase-9 in Human Serum"

_micromachines, 2022, doi:10.3390/mi13101722_

Round 1

Reviewer 1 Report

The authors present a microfluidic chip for protein labeling in the human serum, using two different methods for molds fabrication (micro milling and stereolithography 3D printing), and the experimental characterization of the chip through the labeling of two biomarkers: fibronectin and MMP9. Although the manuscript is of interest, the major concerns that need to be addressed, as described in the topics below, are the short novelty of this manuscript, in comparison to the previous papers of the authors, and the lack of description of the experimental methods (the previous paper is always referred). These topics need to be substantially improved to meet the standards for publication.

1)     The presented work is interesting, but seriously lacks novelty, as similar microsystems have been widely reported in literature. Please explain in detail, and include in the introduction section, the main novelty of this work and where it differs from other published works. In particular, it seems that the previous paper DOI: 10.1007/s00216-022-03915-w already presents the technology, the microfluidic geometry and fabrication, and all the labeling process and measuring with a MR sensor… and this paper only differs by adding also a 3D printed structure for comparison (and even that micromilling / 3D printing comparison is not deeply studied here). How does this work differ from that paper? What interest can it bring to the readers?

2)     Why did the authors select cellular fibronectin and MMP9 as the target biomarkers? Please include a short description of their relevance.

3)     Page 3, line 76: “After the printing process, the mold structure was immersed in isopropyl alcohol (IPA) under an ultrasonic bath”. What is the role of this step?

4)     According to fig 1, D) represents the mold fabricated by 3D printing and C) by micro milling. However, in page 3, line 80, the authors refer: “The fabricated mold from micro-milling (Fig. 1 C and D)”. Clarify/correct it.

5)     In the microchannel presented in fig 1 E), which of the mold fabrication methods was used to obtain that structure?

6)     Include, for instance in fig 1, the dimensions of the microfluidic structure: length, width of the serpentine and chamber, etc…

7)     In fig 2, what was the difference between the regular and optimized micro milling methods? How did the protocols differ? It should be clear in the manuscript. Also, in fig 2 caption, include the microscope magnification.

8)     Still regarding the microfabrication process, in section 3.1, besides table 1, the document lacks a further discussion of the fabrication methods and of the quality of the microchannel structures obtained by micro milling and 3D printing (comparison regarding reproducibility of dimensions, rugosity, angles, etc).

9)     What type of MNPs were used? Material, dimensions, etc? what kind of magnet was used and how was it positioned in the system?

10)  If this paper intends to explore, as suggested in the title, “Labeling on a chip for human serum protein biomarkers”, it must describe with more detail the functionalization and labeling protocols. Referring a previous paper is not enough.

11)  Section 2.3, line 127: What method was used for the immobilization of the monoclonal antibodies into the PDMS substrate?

12)  Section 2.3 – how were the signals acquired and normalized from the MR sensor? Is it a commercial or previously developed sensor? Further details are needed. Add references. Additionally, in line 124, please define it as magnetoresistive (it is the first time that MR appears in the manuscript body).

13)  Section 3.1, line 143 – what reference mold is it? What fabrication process was used? Include a reference.

14)  Statistically, what is the sample size (n) in fig 6?

15)  What are the limits of detection of this biosensor for c-Fn and MMP9?

16)  How does the sensor respond to interference of other biomarkers in the sample? Is it able to still label the targeted biomarker with high specificity? Did the authors study this, for instance, by mixing c-Fn and MMP9 in the same sample?

17)  Overall, the quality of the writing needs improvement. Please revise it. Additionally, found some minor errors. Just a few examples:

- page 1, line 17: in the abstract, something is missing here: “using a simple matrix sample (phosphate buffer.”

- page 2, section 2.1. be coherent regarding the verb tenses (the authors mix, in the same paragraph, simple past, present and future).

- Page 4, line 117 – μL instead of uL

Author Response

Please see the attachment (PDF file named as "Authors Response_Reviewer 1".

Reviewer 2 Report

In the article, “Labelling on a chip for human serum protein biomarker”, the authors have reported a microchip for protein labelling in the human serum. They have utilized CNC machining and SLA-based 3D printing for the fabrication of serpentine microchannel. The article requires extensive revision prior to its acceptance as an archival publication:

Major revisions:

1-The objective of the article is not very clear? Do the authors intend to test which fabrication method (3D printing/ CNC) is more suitable towards microfluidic protein labelling of the human serum?  The reviewer suggests the authors to rewrite the last paragraph of Introduction to highlight the objective and Novelty of this work?

2-Particularly, the reasoning behind the optimized channel height of 150 µm is not scientifically sound. Please provide some experimental or simulation data to validate your claims.

3- The authors should report some of the essential parameters for the biosensing studies, such as cross-reactivity, specificity, and sensitivity of the device etc.?

4-The authors should compare their obtained results to currently available literatures..

5-This articles needs a thorough language editing. The language of this article needs significant improvement.

Minor revisions:

(Abstract): 1-The biomarker labeling 16 occurring in serum was compared using a simple matrix sample (phosphate buffer. Plz remove bracket before phosphate buffer.

2-Figure 1 scale is missing.

3-Figure 6. The legends are missing, x-legend for figure A is overlapped. Please check the right hand legend captions for y-axis

Author Response

Please see the attachment (PDF file named as "Authors Response_Reviewer 2".

Reviewer 3 Report

Manuscript ID micromachines-1920746 entitled „Labeling on a chip for human serum protein biomarkers” describes research on the microfabrication of microfluidic chip for functionalization of biomarkers and magnetic separation of immunocomplexes, along with quality control and application for antigen labeling in spiked serum samples. After careful reading, I have to give a negative comment. The manuscript contains quite little research and the scientific novelty of the developed system is not so eminent - especially in the context of the team's previous work. The manuscript gives the impression that it is only a slightly improved version of an already developed solution (ref. 27). So much so that the manuscript at times is hard to understand without prior knowledge of the authors' previous work (e.g., many elements of the described solution or experimental details). The manuscript is quite short and the optimization studies are presented very superficially (we do not know the number of repetitions, the reproducibility of the manufacturing process, etc.). The conclusions of the results are quite general, and many of them are even obvious. I believe that - although the idea of immunolabeling using microfluidic systems is important and attractive topic - the presented solution is quite poorly characterized and its novelty relative to the authors' previous work is quite small. Therefore, I cannot recommend the manuscript for publication in Micromachines journal. My detailed comments are presented below:   

1)      The title is misleading - the authors do not present an innovative (or at least improved) procedure for labeling of protein biomarkers and the process itself is not the main scientific novelty of the manuscript. The protocol developed and described earlier using functionalized magnetic nanoparticles is only a validation tool for a microsystem fabricated with 3D printing. I recommend that the title be changed to better convey the essence of the research.

2)      The article is virtually unreadable without first reading the ref. [27]. Please provide at least basic and key information about the methodology.

3)      Figure 2. Please clarify exactly what the difference is between non-optimized and optimized micromilling. Was the thickness of the acrylic sheets before micromilling process optimized?

4)      What is the reason for such a large difference in the quality of the images in Fig. 2? If possible, it is worth considering the change of the most blurred one (2A).

5)      The visualization of binding mechanism shown in Figure 2 is unreadable and should be enlarged.

6)      More information on nanoparticles surface modification (type of antibodies used) and assay type should be provided in the Experimental section.

7)      How many repetitions (pieces) were tested during repeatability studies?

8)      The discussion on the effect of channel height on magnetic filtration is supported by the very few results shown in the manuscript and many of the conclusions are quite superficial and obvious (see lines 170-176).

9)      Why do the authors claim that albumin and immunoglobulin G hinder antibody-antigen binding? This statement is quite controversial, because on the other hand, the presence of other proteins promotes the maintenance of antigen stability and antibody biological activity. It would also be worthwhile to test the efficiency of separation from ELISA buffer (e.g. PBST with BSA), typically used as a diluent.

10)   Fig. 6 - The description of the horizontal axis in Figure 6 (upper graph) is obscured.           

Author Response

Please see the attachment (PDF file named as "Authors Response_Reviewer 3".

Round 2

Reviewer 1 Report

The authors addressed my concerns by adding, to the manuscript, explanations and details regarding the experimental methods (the first submission was extremely poor regarding these descriptions). They also clarified the novelty of this work when compared to their previous paper, focusing on the labeling medium (which was not previously clear). Also, the title was improved and describes better the presented work. The paper was significantly improved.

Author Response

We appreciate the recognition of our efforts in improving our manuscript, and we thank you for the insightful, careful reading, and important comments that allowed us to improve it. We highlight that the abstract has been revised again to emphasize the main novelty of the study carried out with a serum-based matrix.

Reviewer 2 Report

My questions are addressed by the authors. The paper can now be accepted for publication.

Author Response

We appreciate the recognition of our efforts in improving our manuscript, and we thank you for the important comments that allowed us to improve it. We highlight that the abstract has been revised again to emphasize the main novelty of the study carried out with a serum-based matrix.

Reviewer 3 Report

Dear Authors,

I thank you and appreciate the work you have put into improving and developing most of the topics raised in the review (although a few of them have been skimmed rather evasively). However, due to the nature of the objections (primarily concerning the low scientific novelty and low originality with respect to previous, already published works - which were also recognized by another Reviewer), I cannot change my opinion.

Author Response

We appreciate the recognition of our efforts in improving our manuscript, and we thank you for the important comments that allowed us to improve it.